# Point-process modeling of secondary crashes

**Samarth Motagi[1], Sirish Namilae[1]\*, Audrey Gbaguidi[1], Scott Parr[2], Dahai Liu[3]**

**1** Department of Aerospace Engineering, Embry-Riddle Aeronautical University, Daytona Beach, Florida, United States of America, **2** Department of Civil Engineering, Embry-Riddle Aeronautical University, Daytona Beach, Florida, United States of America, **3** College of Aviation, Embry-Riddle Aeronautical University, Daytona Beach, Florida, United States of America

\* namilaes@erau.edu

**Data Availability Statement:** All relevant data are within the manuscript and supporting information files.

**Funding:** This research was funded by DOT-UTC Center for Advanced Transportation Mobility. The funders had no role in study design, data collection

## Abstract

Secondary crashes or crashes that occur in the wake of a preceding or primary crash are among the most critical incidents occurring on highways, due to the exceptional danger they present to the first responders and victims of the primary crash. In this work, we developed a self-exciting temporal point process to analyze crash events data and classify it into primary and secondary crashes. Our model uses a self-exciting function to describe secondary crashes while primary crashes are modeled using a background rate function. We fit the model to crash incidents data from the Florida Department of Transportation, on Interstate-4 (I-4) highway for the years 2015–2017, to determine the model parameters. These are used to estimate the probability that a given crash is secondary crash and to find queue times. To represent the periodically varying traffic levels and crash incidents, we model the background rate, as a stationary function, a sinusoidal non-stationary function, and a piecewise non-stationary function. We show that the sinusoidal non-stationary background rate fits the traffic data better and replicates the daily and weekly peaks in crash events due to traffic rush hours. Secondary crashes are found to account for up to 15.09% of traffic incidents, depending on the city on the I-4 Highway.

## Introduction

Highway crashes can have an immediate and significant impact on the safety of individuals and the mobility of goods. According to National Highway Traffic Safety Administration (NHTSA), motor vehicle accidents resulted in over 42,939 and 42,795 deaths in the year 2021 and 2022 respectively in the United States [1]. Drivers near the crash area must react rapidly to a dynamic and unpredictable environment in the immediate aftermath of a crash. Vehicles appear to queue up on the highway route as they reach the crash site. Furthermore, the crash scene itself is a distraction to drivers in both directions. This situation can increase the likelihood of yet another crash.

Secondary crashes are incidents that occur as a result of a primary accident. According to estimates, almost ten percent of highway crashes are categorized as secondary [2]. The victims of the primary crash, as well as the first responders dispatched to assist them, are in grave danger in these secondary crashes. Many organizations have missions and forums that encourage

and analysis, decision to publish, or preparation of
the manuscript.

**Competing interests:** There are no competing
interests for this study.

drivers and emergency responders to learn about the value of "move over" laws for protecting
individuals employed on the side of the road. Comprehending why secondary crashes occur
and predicting where and when they occur will help protect vulnerable road users, including
primary crash victims and emergency responders.

Given the difficulties in modeling secondary crashes, we plan to utilize a new modeling
approach to study secondary crashes based on a point process model. Secondary crashes
exhibit characteristics of social contagion. The traffic and road conditions developed during
the primary crash lead to secondary crashes in a manner similar to contacts with an infected
individual leading to further disease spread. Such diffusion of events or information from a
primary event has been studied using Hawkes point process models [2–6]. We have recently
utilized epidemiological models to study the spread of fuel shortages during hurricane evacua-
tions [7]. These studies point to the success of epidemiological models in examining the
dynamics of social contagion.

We propose to utilize a self-exciting point process model commonly used in the study of
earthquakes [8], known as Epidemic Type After Shock (ETAS) model to quantify secondary
crashes from a traffic dataset. The self-exciting point process models are commonly used to
classify the dataset of discrete events into background and offspring events. For example, in
case of earthquakes, the background events are the independent mainshock events, while the
dependent aftershocks are considered to be secondary events. While there have been several
modeling advances, and code developments focusing on seismic modeling [8–12], several
researchers have applied this concept to other clustered societal problems. For example, Moh-
ler et al. [2] used the self-exciting point process model to understand crime events; Zhao et al.
[12] used a self-exciting point process model to predict tweet popularity by modeling informa-
tion cascades. Dassios et al. [13] modeled the contagion risks or clustering events in finance
and insurance while Bertozzi et al. [14] used Hawke's point process models to understand the
email communication between individuals in an organization. Towers et al. [6] employed the
same approach to identify the contagion in mass killings and school shootings. Lewis et al. [15]
characterized the temporal patterns of violent civilian deaths in Iraq using a self-exciting point
process model.

Recent research has assessed the influence of a primary highway crash on the occurrence of
subsequent secondary crashes through the utilization of a Zero Inflated Ordered Probit
(ZIOP) regression model [13]. This approach incorporates real-time traffic conditions to
establish connections between the likelihood of multiple secondary crashes and various factors
such as current traffic flow, geometric attributes, weather conditions, and characteristics of the
initial crash following a primary collision. Kitali et al. [16] used real-time speed data from
BlueToad paired devices to identify secondary crashes. Sarker et al. [17] proposed Generalized
Ordered Response Probit (GORP) models that would allow us to predict the frequency of sec-
ondary crashes occurrence based on segment and traffic characteristics. Zhang et al. [18]
developed a network-based clustering algorithm based on crowdsourced Waze user reports
from the primary crash, with any subsequent crashes occurring inside the cluster of primary
crash is considered to be secondary crash. Salek et al. [19] presents a method for assessing the
likelihood of freeway secondary crashes with Adaptive Signal Control Systems (ASCS)
deployed on alternate routes. Song et al. [20] present a methodology using link-based speed
data to classify all reported crashes based on operational conditions, without prior identifica-
tion of the cause. A case study on a 274 km section of I-40 revealed 12% of crashes in non-
recurrent congestion, with 37% linked to unreported primary incidents, while the rest were
classified as primary crashes in uncongested conditions (84%) or recurrent congestion (4%).
The methodology is easily implementable in any advanced traffic management system with
crash time, link location, and archived link speed data. Zhang et al. [21] use a binary logit

model to identify the contributing factors for secondary crashes on Utah highways. They develop a hierarchical ordered probit (HOPIT) model for analyzing injury patterns in primary and secondary crashes. Another recent study [22], employs machine learning techniques for forecasting the likelihood of crashes on freeways based on traffic patterns just prior to accidents, including factors like average speed and speed reductions, in combination with freeway characteristics.

In this paper, a novel approach for identifying secondary crashes using the self-exciting point process model is formulated. To account for periodic variations in traffic events, corresponding periodic background rate variations are introduced. The model is employed to analyze the crash events on a dataset corresponding to Interstate Highway-4 (I-4) in Florida, USA. The results from the point process model compare favorably with other models used to analyze secondary crashes. This approach provides a new statistical tool for analyzing this ubiquitous transportation problem.

## Methodology

### Traffic and crash data

The crash data for this study is obtained from the Florida Department of Transportation (FDOT) Safety Office's GIS Query Tool (SSOGIS) on Interstate4 (I-4) for the years 2015–2017 [23]. The I-4 highway between Daytona Beach, FL and Tampa, FL is entirely located in the state of Florida, USA. All the crashes on this highway are listed in the FDOT Crash Analysis and Reporting database. The data reports several crash characteristics including crash date, crash time, longitude, latitude, weather condition, crash location, speed limit, type of injury, light condition, and surface condition of the road. This data does not identify if the crash events are primary or secondary.

The crash dataset used in this research contains information on 6367, 6663 and 6133 crash events on I-4 for the years 2015, 2016 and 2017 respectively. A temporal distribution of the crash data indicates that the highest number of crashes occurred during the evening rush-hour, followed by morning rush-hour. Also, the highest number of crashes occurred on Fridays, and the lowest on Sundays. The histograms in Fig 1 show those variations. The FDOT crash data did not specify which of the incidents were primary and which were secondary crashes.

To validate our model's findings, the I-95 freeway crash dataset that was analyzed by data-driven approach in [24,25] is obtained and compared with the point-process model developed in this paper. The I-95 highway for the study is considered from Jacksonville to Miami which is in the state of Florida. The crash dataset we considered consists of 887 crashes in Miami and 309 crashes in Jacksonville from January 2017 to June 2019.

### Point process model formulation

In a self-excitation point process model, recent prior events increase the probability of another event happening in the near future. Primary crashes are often sites for reduced mobility which causes a shock wave in traffic flow. The shock waves are the byproduct of traffic congestion and queuing [26]. When a driver enters through a shockwave, one experiences a sudden change in vehicle speed, which often causes secondary accidents. Secondary accidents are also caused due to rubbernecking phenomena as the driver approaches the vicinity of the crash location site. Given a time crash database consisting of N crash events $(t_i)$, i = 1, . . ., N, representing respectively the time of the i$^{th}$ crash event, the conditional intensity function is given by:

$$\lambda(t) = \mu + \sum_{i:t_i<t} g_{A,\alpha}(t - t_i) \tag{1}$$

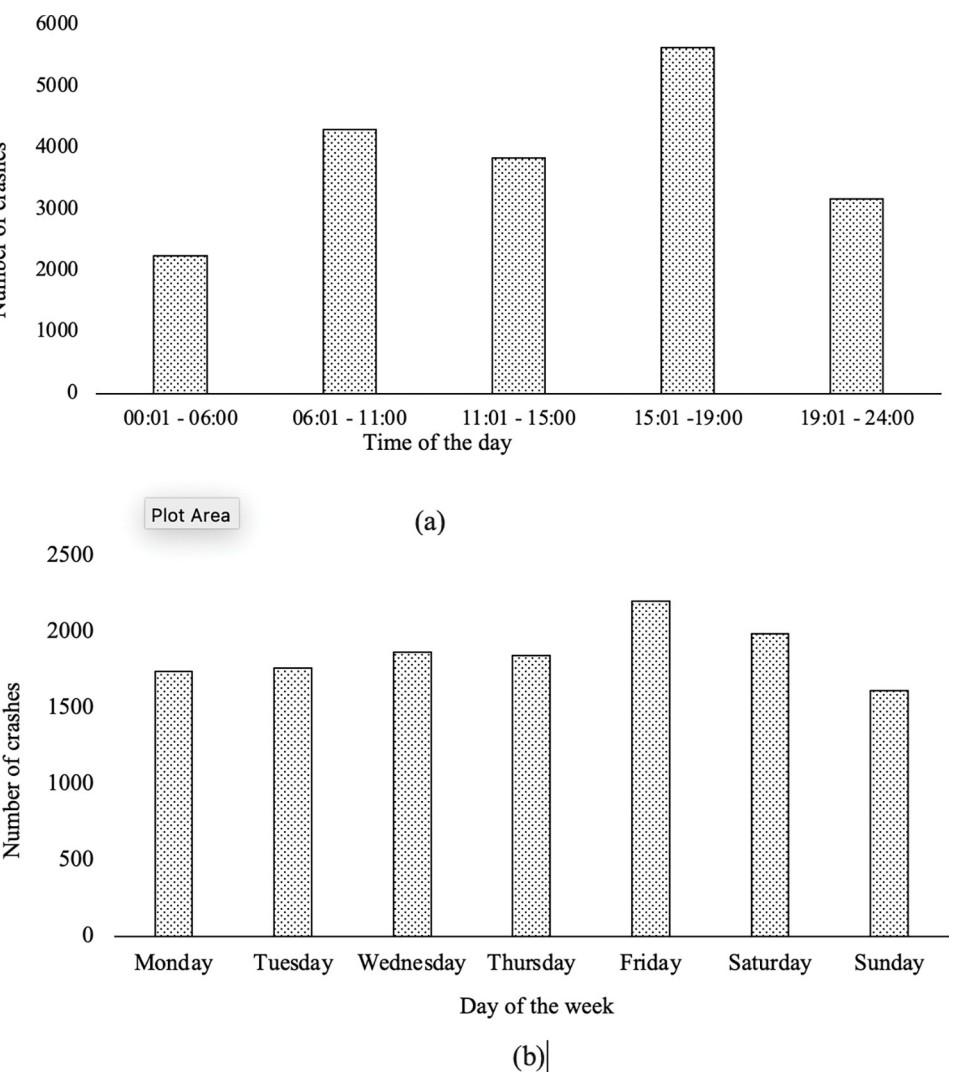

**Fig 1. Histograms showing the distribution of crashes according to (a) time and (b) day of the week.**

$\lambda$ is the limiting expected rate of crash events, given the history $H_t = \{(t_i) : t_i < t\}$ of all crash events in a region S, up to time t. The model classifies crash events in the catalog data into two categories, primary crashes and secondary crashes. Primary crashes are assumed to occur independent of time at a rate $\mu > 0$.

$g_{A,\alpha}(t - t_i)$ is the probability density function (PDF) of the occurrence time of a secondary crash at time $t_i$ and is modeled as

$$g_{A,\alpha}(t - t_i) = \begin{cases} A * \alpha * exp[-\alpha * (t - t_i)], t > t_i \\ 0 \qquad\qquad\qquad\qquad , t \le t_i \end{cases} \quad (2)$$

The conditional intensity function $\lambda(t|H_t)$ can now be expressed as,

$$\lambda(t) = \mu + A \sum_{i:t<t} \alpha \cdot e^{-\alpha(t-t_i)} \quad (3)$$

Here, $\alpha$ indicates the decay rate in time for the occurrence of secondary crashes while $A$ represents the amount of excitation generated by the collection of prior events. Note that $g_{A,\alpha}$ is the function describing the time lag $t-t_i$ between a secondary crash and corresponding primary crash. The model is function of three unknown parameters $\theta = (\mu, A, \alpha)$. These parameters are determined through maximum likelihood estimation while fitting the model to the crash data. Given a time sequence of crash data consisting of N crash events $\{(t_i), i = 1, \ldots, N\}$, during a time interval $[0, T]$ and in a region S, the log-likelihood function [24] is:

$$l(\theta|H_t) = \sum_{i=1}^{N} log\lambda_\theta(t_i|H_{ti}) - \int_0^T \iint_S \lambda_\theta(t|H_t)dt \tag{4}$$

To obtain the maximum likelihood estimate (MLE) of $\theta$, the Davidon-Fletcher-Powell method is used as a gradient-based nonlinear optimization procedure [24]. A numerical approximation of the integral terms in the log-likelihood function is first computed. A probabilistic approach previously used for declustering an earthquake catalog [9] is used here to decluster the crash data and to obtain the primary crash rate $\mu$. The probability that crash event 'i' triggered crash event 'j' is:

$$p_{ij} = \begin{cases} \dfrac{g_{A,\alpha}(t_j - t_i)}{\lambda(t_j|H_{t_j})} & , t_j > t_i \\ 0 & , t_j \leq t_i \end{cases} \tag{5}$$

Hence the probability of crash event j being a secondary crash is $p_j = \sum_{i=1}^{j-1} p_{ij}$. Consequently, the probability of crash event j being a primary crash is:

$$1 - p_j = \frac{\mu}{\lambda(t_j|H_{t_j})} \tag{6}$$

Once the MLE of $\theta$ is found and the probability values $p_j$ of all N crash events are found, the crash events can be classified into primary crashes and their related secondary crashes. The maximum queue time for secondary crashes can then be extracted from the model. The queue time is the maximum time lag between primary crashes, and their related secondary crashes.

## Periodic variation of the background function

The model in Eq (3) assumes a constant background rate. However, the crash data varies periodically on a weekly and daily basis, as shown in Fig 1. To examine this variation, a non-stationary background rate is introduced. Other researchers have used a similar approach. For example, when Lewis et al. [15] inspected the data on violent death in Iraq using a stationary background rate, the results failed to capture the upward trend over time; that issue was addressed using a non-stationary background rate model. Fox et al. [27] used a nonstationary background rate to investigate daily and weekly email activity trends.

To capture the periodic traffic and crash variations through the background rate, we use two functions: (a) a step function and (b) a sinusoidal function. A step function is used to model piecewise variation in background rate. This allows the background rate to jump to different stationary levels at different time periods. The conditional intensity function $\lambda(t|H_t)$ with non-stationary background rate as a step function can be expressed as:

$$\lambda(t) = \mu step(t) + A \sum_{i:t_i<t} \alpha.e^{-\alpha(t-t_i)} \tag{7}$$

Here, when modeling the variation in crash rates for different days of the week (see Fig 1 (b)), the background rate is expressed as:

$\mu_{step}(t) = \mu_i$ for i = 1 to 7, representing the 7 days of the week.

Similarly, when modeling the rush-hour vs non rush hour traffic:

$\mu_{step}(t) = \mu_i$ for i = 1 to 3, representing morning rush-hour, non-rush-hour and evening rush-hour.

Alternatively, to account for weekly and daily traffic periodicity, a sinusoidal function is used to represent the non-stationary background rate. The conditional intensity function $\lambda(t|H_t)$ can then be expressed as:

$$\lambda(t) = \mu(t) + A \sum_{i:t_i < t} \alpha \cdot e^{-\alpha(t-t_i)} \tag{8}$$

Here, the best fit function for the weekly trend can be expressed as:

$$\mu(t) = \mu_{\sin}(t) = \mu_o \times \{P \times \sin(Q \times t + R) + S\} \tag{9}$$

Similarly, when modeling the rush-hour vs non-rush-hour, daily variation in the background function is expressed as:

$$\mu(t) = \mu\cos(t) = \mu_o \times \{P_h \times \cos(Q_h \times t + R_h) + S_h\} \tag{10}$$

Here, P, Q, R, S and $P_h$, $Q_h$, $R_h$, $S_h$ are the parameters for background rate that fit the weekly and daily crash trend respectively. These parameters describe the sinusoidal variation of the crash data and are obtained using the data shown in Fig 1(a) and 1(b).

The R code for the ETAS model obtained from Jalilian [10] is adapted to identify the secondary crash events in the present work. The code was developed to analyze an earthquake catalog using the stochastic declustering approach. We modified the code for a temporal-only self-exciting point process for the current application. The input data contains the date and time of the crash events. We also utilized a separate MATLAB code for post-processing to classify potential secondary crashes from the dataset based on queue time and corresponding probability values.

## Results

### Model validation

Detailed validation is performed by comparing our methodology with a recent data-driven approach in the literature [24,25]. In this approach, BlueToad® speed records are used in a data-driven methodology to identify secondary crashes. Salum et al. [24] utilized this approach to analyze secondary crashes on Florida highways and investigated the safety benefits of the Freeway Service Patrol program, Road Rangers, for preventing secondary crashes [24], and reducing the clearance duration [25].

We utilized an I-95 highway crash dataset analyzed using the approach in [24] and compared it with the point-process model developed in this study. We investigated two cities on I-95, Jacksonville, and Miami, considering a 20-mile length of the highway for each city. We set a spatial threshold of two miles for each simulation. The simulations are repeated by changing the region of interest by 0.25 miles from the previous start point for 20 miles length for each city. The data was modeled using a constant background rate and a sinusoidal background rate. The model parameters were optimized using a maximum likelihood estimate. In Eq (3), A is the measure of the excitation generated from previous events, and α represents the rate of decay in the triggering function. The inverse of parameter α in the triggering function for time

**Table 1. Validation results for Miami and Jacksonville on I-95 freeway.**

| Parameters | Miami | Jacksonville |
|---|---|---|
| Total number of crashes | 887 | 309 |
| Number of secondary crashes by data-driven technique [24] | 52 | 27 |
| Number of secondary crashes using constant background rate that match with literature in [24] | 46 (88.4%) | 24 (88.8%) |
| Number of secondary crashes using sinusoidal background rate that match with literature in [24] | 48 (92.3%) | 25 (92.6%) |
| Average probability of a secondary crashes | 0.89 | 0.95 |

(g) provides the average time window over which secondary crash can occur following a primary crash. Validation results for Miami and Jacksonville area on I-95 freeway is shown in Table 1. The average time for secondary event for this data was 52 minutes for Jacksonville and 38 minutes for Miami for stationary background rate. The corresponding values for sinusoidal background rate are 43 minutes and 33.4 minutes respectively. This implies that any crash that occurs within the queue time and the 2-mile spatial threshold of the primary crash can be categorized as a secondary crash. Of the 887 collisions in the area of interest in Miami, 52 were identified as secondary crashes using a data-driven technique [24]. This data was also analyzed using methods developed in the current paper using constant background rate and the results showed that 46 of the 52 crashes (88.4%) mentioned above were identified as secondary crashes. The temporal point-process model with sinusoidal background rate estimated that 48 of the 52 crashes (92.3%) mentioned above were identified as secondary crashes. Note that we also calculate the probability that a given crash is a secondary crash. The aforementioned 48 crashes have an average probability of 0.89 to be secondary crashes. In addition, our approach identified 10 additional crashes as potential secondary crashes, with a lower probability (0.77). The four crashes which were identified only in the data-driven approach had a very high temporal threshold. Similarly, for Jacksonville, there were 309 crashes in the area of interest. Of these, 27 were identified as secondary crashes using the data-driven technique [22]. Using the temporal point-process with a constant background rate, 24 of the 27 crashes (88.8%) were identified as secondary crashes, and 25 of the 27 crashes (92.6%) were identified as secondary crashes using sinusoidal background rate. These 25 crashes have an average probability of 0.95 to be secondary crashes. In addition, our approach identified four additional crashes as potential secondary crashes, with a lower probability.

## Model application to interstate I-4 crash data

The proposed approach is demonstrated on a dataset consisting of all crashes on Interstate-Highway-4 (I4) in Florida, USA for three years (2015 to 2017) [23]. The interstate highway I-4 is located entirely within the state of Florida, spanning 132.3 miles [28]. In the west, the I-4 begins in Tampa and ends in the east in Daytona Beach. The model is investigated in six different cities on I-4: Tampa, Plant City, Kissimmee, Orlando, Sanford, and Daytona Beach. Tampa and Orlando are major cities and are chosen because their corresponding data contains more than 500 crash events. The other cities modeled here have at least 200 crash events.

Spatial and temporal thresholds from a primary incident are often used to analyze secondary crashes. Recent studies by multiple investigators [29–31] indicate that the spatial threshold is about 2-miles. Therefore, we analyze the data within a threshold of 2-miles of the highway for the central parts of the above-mentioned cities. Furthermore, our spatial scope encompasses crashes both downstream and upstream within the 2-mile radius, contributing to a comprehensive evaluation of the crash dynamics surrounding primary crash. Initially, the

**Table 2. Estimated parameter values, no of secondary crashes and average time of secondary event represented for various cities on I-4 for constant background rate.**

| City | Total crashes | mu (μ) | A | Alpha (α) | Secondary crashes | Secondary event avg. time (min) |
|---|---|---|---|---|---|---|
| Tampa | 613 | 0.495 | 0.114 | 10.459 | 69 (11.25%) | 137.7 |
| Plant city | 322 | 0.259 | 0.115 | 22.375 | 27 (8.38%) | 64.3 |
| Kissimmee | 207 | 0.178 | 0.052 | 15.323 | 10 (4.83%) | 93.9 |
| Orlando | 583 | 0.392 | 0.262 | 12.465 | 121 (20.75%) | 115.5 |
| Sanford | 248 | 0.205 | 0.090 | 8.676 | 19 (7.66%) | 166.0 |
| Daytona Beach | 242 | 0.190 | 0.138 | 10.401 | 25 (10.33%) | 138.4 |
| **Average** | | | | | | **119.3** |

model is implemented with a constant background rate. It is then used to examine the periodic variation of weekly and daily trends by employing non-stationary background rates using sinusoidal and piecewise functions. Finally, the Akaike Information Criterion (AIC) [32] is used to identify the best-fit models.

## Stationary background rate

Table 2 shows the optimized parameters for the constant background rate model, based on the maximum likelihood estimate. Here, *A* is the measure of excitation generated from the previous events and, $\alpha$ represents the rate of decay in the triggering function in Eq (3). Based on the functional form of the triggering function (*g*), the reciprocal of $\alpha$ gives the average time-window over which secondary crash can occur following a primary crash. For example, from Table 2, in case of Tampa $\alpha^{-1}$ = 0.0956 days = 137.7 minutes. This implies that if a crash occurs within 137.7 minutes and a 2-mile distance from the primary crash location, the crash is related to the primary crash and can be classified as secondary crash. The queue time evaluated from the model is used to classify the crash dataset into the primary and secondary crashes. The classification is conducted based on the probability values obtained using Eq (6). The number of secondary crashes shown in Table 2 is calculated using a post-processing code. The post-processing code finds the time difference between successive events, and if the event with corresponding time difference is lesser than the queue time and corresponding threshold probability, the event is considered a secondary crash. The average queue time of secondary crash events for all the cities studied here is 119.3 minutes for the stationary background rate. Fig 2 shows the primary and the secondary crash locations for a two-mile stretch for Tampa, Orlando, and Daytona Beach.

## Non-stationary background rate

The point process model with a non-stationary background rate using a piecewise function is examined next. Here the background rate is set to vary at different stationary levels on different days in a week as described earlier. Fig 1(b) shows that the number of crashes on Friday is 136% of that on Sunday. Similarly, a non-stationary background point process model using a sinusoidal function shown in Eqs 8 and 9 is simulated to capture this periodic variation. Table 3 shows the estimated parameters for the cities considered for the piecewise background rate and sinusoidal background rate variations, respectively. MLE based optimization is again used to estimate the parameters. The queue time reported in the Table 3 is a weighted average of the values for the seven days of the week. Once the average queue time is calculated for each city, the number of secondary crashes is evaluated using the post-processing described earlier.

Similar piecewise and sinusoidal functions for background rate variation are used to examine the periodic variation in daily crash trends for rush-hour and other periods as reported in

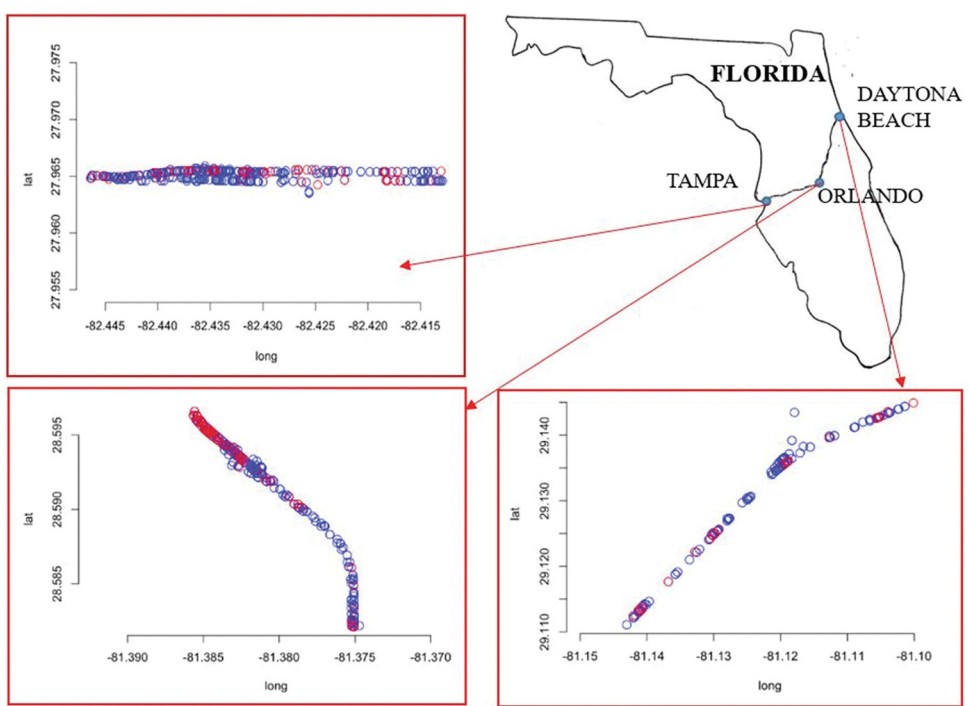

**Fig 2. Primary and secondary crashes classified from the model for Tampa, Orlando and Daytona Beach along the I-4 highway.** Secondary crashes are shown in red.

Fig 1(a). These are described in Eqs 8 and 10. Table 4 shows the corresponding number of secondary crashes and average queue times. The queue time varies for each time-period and the average queue time shown in Table 4 is a weighted average value for rush-hour and other time-periods.

Fig 3 shows the variation of queue time for different time-periods for the Orlando region. Here the queue time is higher for the morning rush hour period and the evening rush hour

**Table 3. Estimated parameter values, no of secondary crashes and average time of secondary event represented for various cities on I-4 for piecewise and sinusoidal background rate for the different days of the week.**

| Piecewise | City | Total crashes | mu ($\mu$) | A | Alpha ($\alpha$) | secondary crashes | Secondary event avg. time (min) |
|---|---|---|---|---|---|---|---|
| | Tampa | 613 | 0.491 | 0.121 | 9.511 | 61 (9.95%) | 121.1 |
| | Plant city | 322 | 0.262 | 0.105 | 26.378 | 23 (7.14%) | 46.1 |
| | Kissimmee | 207 | 0.179 | 0.050 | 16.025 | 8 (3.86%) | 75.8 |
| | Orlando | 583 | 0.405 | 0.238 | 15.411 | 101 (17.32%) | 78.9 |
| | Sanford | 248 | 0.206 | 0.089 | 8.715 | 18 (7.25%) | 139.6 |
| | Daytona Beach | 242 | 0.190 | 0.138 | 10.412 | 24 (9.91%) | 116.8 |
| | **Average** | | | | | | **96.3** |
| Sinusoidal | Tampa | 613 | 0.588 | 0.089 | 14.706 | 52 (8.48%) | 82.7 |
| | Plant city | 322 | 0.305 | 0.101 | 27.664 | 22 (6.83%) | 43.9 |
| | Kissimmee | 207 | 0.208 | 0.046 | 16.466 | 7 (3.38%) | 73.8 |
| | Orlando | 583 | 0.476 | 0.225 | 17.073 | 87 (14.92%) | 71.2 |
| | Sanford | 248 | 0.241 | 0.075 | 10.951 | 15 (6.04%) | 112.8 |
| | Daytona Beach | 242 | 0.220 | 0.134 | 10.933 | 23 (9.50%) | 110.4 |
| | **Average** | | | | | | **82.4** |

**Table 4. No of secondary crashes and average time of secondary event for piecewise and sinusoidal background rates, considering rush-hour and non-rush hour variations.**

| City | Piecewise background rate | | Sinusoidal background rate | |
|---|---|---|---|---|
| | Secondary crashes | Secondary event avg. time (min) | Secondary crashes | Secondary event avg. time (min) |
| Tampa | 53 (8.64%) | 96.1 | 51 (8.31%) | 88.2 |
| Plant city | 22 (6.83%) | 36.6 | 22 (6.83%) | 37.5 |
| Kissimmee | 7 (3.38%) | 66.7 | 6 (2.89%) | 57.9 |
| Orlando | 87 (14.92%) | 67.0 | 81 (13.89%) | 60.1 |
| Sanford | 15 (6.04%) | 109.6 | 15 (6.04%) | 95.1 |
| Daytona Beach | 21 (8.67%) | 92.6 | 21 (8.67%) | 86.4 |
| **Average** | | **78.1** | | **70.8** |

periods. The higher traffic rates during rush-hour [33] lead to an increase in the number of crashes at these times as shown in Fig 1(a). This in turn leads to higher queue times during rush hour. The underlying data for parametrizing Eqs (8) and (9) is based on Fig 1(a), therefore, the queue time follows the same trend for both sinusoidal and piecewise background rate variations. The results for the data corresponding to other cities are similar to Fig 3.

## Model comparison

Akaike Information Criterion (AIC) [32] is used to find the goodness-of-fit for the proposed models. The AIC estimates the relative amount of data loss when a statistical model is fit to a given dataset. Given a set of candidate models, the model with a relatively lower AIC value is considered to be the best fit. The AIC is given by:

$$AIC = 2k - 2\ln(L) \qquad (11)$$

Where $k$ is the number of model parameters and, $L$ represents the maximum value of the

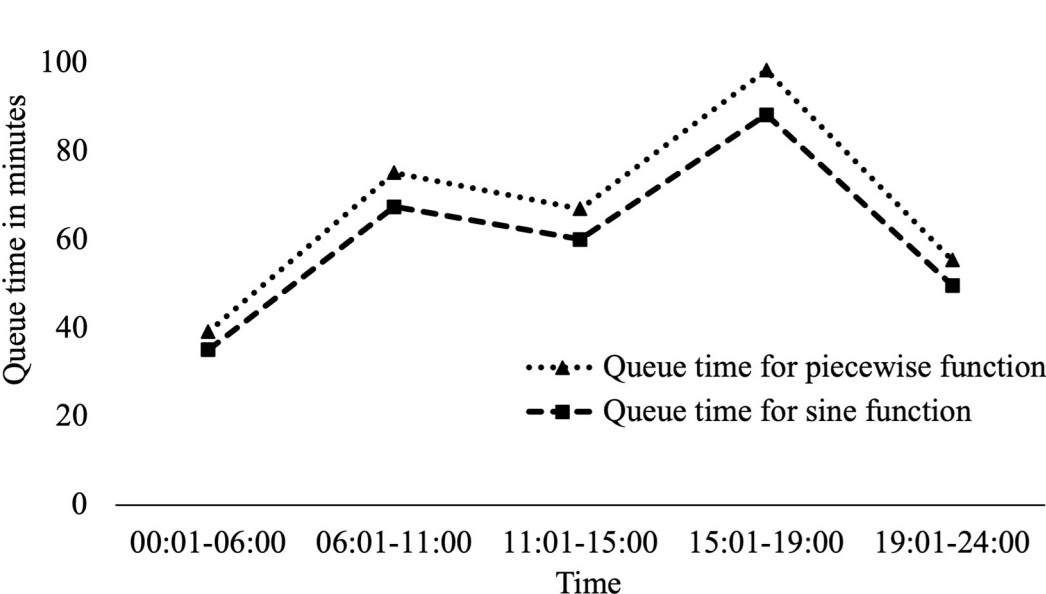

**Fig 3. Queue time for Orlando region during different time of the day.**

**Table 5. Comparison of models using AIC values.**

| | AIC values for stationary background rate | AIC values for weekly trend | | AIC values for daily trend | |
|---|---|---|---|---|---|
| Cities | $\mu$ | $\mu_{step}(t)$ | $\mu_{sin}(t)$ | $\mu_{step}(t)$ | $\mu_{cos}(t)$ |
| Tampa | 2037.24 | 2010.92 | 1894.67 | 2043.00 | 2001.81 |
| Plant city | 1570.23 | 1417.07 | 1381.89 | 1446.74 | 1424.88 |
| Kissimmee | 1185.62 | 1131.92 | 1099.97 | 1147.41 | 1125.74 |
| Orlando | 1944.56 | 1710.30 | 1638.90 | 1768.66 | 1720.58 |
| Sanford | 1289.25 | 1257.19 | 1218.55 | 1278.02 | 1253.69 |
| Daytona Beach | 1198.58 | 1192.69 | 1168.87 | 1178.41 | 1163.56 |

likelihood function for the model. While comparing the models with a different number of parameters, a penalty of *2k* is added to reject any overfitting by the model. The AIC values for each model, with stationary and non-stationary background rates for weekly and daily traffic trends for each city, are shown in Table 5. The AIC values for stationary background rate are higher compared to non-stationary background rate. This suggests that the stationary background rate model has a relatively higher prediction error compared to other models. Further, the sinusoidal background rate models outperform the piecewise function. Similarly, the sinusoidal model exhibits lowest AIC for the hourly variation as well.

## Discussion

Several studies listed in Table 6 model secondary crashes using different approaches. The key parameters in these studies are the temporal and spatial boundaries of primary incidents that define secondary crashes. The studies based on the static method [29,31,34,35] model fixed and predetermined spatial and temporal thresholds for secondary crash identification. However, these studies often do not yield a uniform threshold for secondary crashes since the traffic conditions, geometric characteristics, and incident attributes vary for each crash. Various dynamic methods [36] using varying spatio-temporal thresholds were proposed in recent studies to improve the identification of secondary crashes. In the context of the current research,

**Table 6. Literature review on identifying secondary crashes.**

| Authors | Method | Spatio-temporal threshold |
|---|---|---|
| Raub [34], Karlaftis et al [37]. | Static | 1-mile and 15 minutes |
| Moore et al [29]. | Static | 2-mile and 2 hours; same for opposite direction |
| Hirunyanitiwattana et al [30]. | Static | 2-mile and 1 hour |
| Zhan et al [31]. | Static | 2-mile and 15 minutes |
| Chang et al. [38] | Static | 2-mile, 2 hours, and 0.5-mile, 0.5 hour (opposite direction) |
| Sun et al. [36] | Dynamic | Incident progression curve |
| Zhang et al. [39] | Dynamic | Queue length estimations |
| Zhan et al. [40] | Dynamic | Cumulative arrival and departure plots |
| Chou et al. [41] | Dynamic | Simulation-based methods |
| Yang et al. [42], Xu et al. [43], Kitali et al. [16], Chung et al. [44] | Dynamic | Spatio-temporal impact area methods based on speed contour plot |
| Imprialou et al. [45] | Dynamic | Automatic tracking of moving jams |
| Park et al. [46] | Dynamic | Vehicle probe data |
| Sarker et al. [47], Mishra et al. [48], Junhua et al. [49], Wang et al. [50] | Dynamic | Shockwave principles |

**Table 7. Comparison of average time of secondary crash event for each city on I-4.**

| City | Average queue time (minutes) and % secondary crashes | | |
|---|---|---|---|
| | Constant background rate | Piecewise background rate | Sinusoidal background rate |
| Tampa | 137.7 (10.44%) | 127.9 (9.78%) | 82.7 (8.15%) |
| Plant city | 64.3 (8.38%) | 46.1 (7.14%) | 43.9 (7.14%) |
| Kissimmee | 93.9 (4.34%) | 75.8 (3.86%) | 73.8 (3.38%) |
| Orlando | 115.5 (19.55%) | 78.9 (16.29%) | 71.2 (15.09%) |
| Sanford | 166.0 (7.66%) | 139.6 (7.25%) | 112.8 (6.45%) |
| Daytona Beach | 138.4 (10.33%) | 116.8 (9.50%) | 110.4 (9.09%) |
| **Average** | **119.3** | **97.5** | **82.4** |

the stationary background rate approach in Eq 3 can be compared to the static methods. In contrast, the model with a non-stationary background rate for piecewise and sinusoidal functions is similar to the dynamic approaches.

The queue time calculated from the three models is compared in Table 7. The average queue time of 119.3 minutes is obtained for stationary background rate. The AIC values are higher for the model with a stationary background rate than the other two models as it fails to capture the periodic variation of the crash trend. Average queue time of 97.5 minutes is obtained for the non-stationary background rate model using the piecewise function. For sinusoidal function a queue time of 82.5 minutes is obtained. The AIC values are the least for sinusoidal function compared to the other two models. These queue time values are comparable to the literature in Table 6.

The secondary crash percentage from the models with stationary and non-stationary background rates are shown in Table 7. In general, the percentage of secondary crashes as well as queue time obtained from the stationary background rate model are higher than those obtained with models with non-stationary background rates. Recent studies show a similar percentage of secondary crashes with 8.8% in Moore et al. [29]; 7.5% in Chung [44]; 6–7% in Kalair et al. [51]; 8.42% in Yang et al. [52]; 5.22% in Zhan et al. [31]; 5.53% in Kopitch et al. [35]; 3.23% in Zhan et al. [40] for different urban locations. The results from the current model are comparable to the literature. The point process model provides an alternate reliable modeling approach for classifying secondary crashes and devising policies and procedures to reduce the resulting loss of life and property damage.

The percentage of the secondary crash for the Orlando region is higher compared to other cities in the study. There are likely a number of factors which contribute to this phenomenon including, construction and maintenance, unique lane geometry of the Orlando section, high traffic volumes, and urban setting. However, one factor that is unique to Orlando and likely plays a significant role in the secondary crashes is the prevalence of unfamiliar drivers. Orlando is one of the top tourist destinations in the USA, receiving approximately 75 million domestic and international visitors annually. Unfamiliar drivers tend to be more distracted, as they seek route guidance via roadside signs or GPS system. Such conditions likely contribute to secondary crashes.

Traffic incident management (TIM) programs are designed to detect, respond to, and clear traffic incident scenes. These multi-disciplinary efforts aim to treat victims, clear vehicle wreckage, and restore the flow of traffic safely and quickly. Effective TIM programs reduce incident duration and affect, while improving safety for crash victims, the motoring public, and emergency responders. The adoption of a planned and coordinated approach to protect the incident scene, modify the flow of traffic, and separate crash victims and responders from the motorist may have a profound impact on the severity and frequency of secondary crashes [53]. The results of this research including probability distribution of secondary crashes and the secondary event average time could inform TIM toward a targeted clearance time.

Decreasing the time needed to clear a primary incident reduces the exposure of the scene to secondary crashes. This research found that secondary crashes, on average occurred between 37 and 96 minutes after the primary incident, when accounting for the natural fluctuations in traffic and incident patterns. This may suggest that reducing the primary incident impact duration below this threshold, could significantly reduce the likelihood of secondary crashes. While the model is parameterized to a specific highway in this work, the approach is generic and can be used for the any crash and traffic datasets.

The utilization of Connected and Autonomous Vehicle (CAV) technology on American freeways is currently in its initial phases. Pilot projects are aiding in the collection of data and the testing of the safety and dependability of CAV technology. As this technology advances, it is foreseeable that an increased number of CAVs will operate on US freeways. With the emergence of connected vehicle (CV) technologies, it's highly probable that CVs will soon possess the capability to communicate with each other using wireless vehicular network. This information exchange among vehicles is anticipated to alter traffic operations and empower drivers to take more proactive measures. Rahman et al. [54] conducted a study to assess the safety of CV platoons by comparing managed-lane CV platoons and all-lanes CV platoons. The study's findings demonstrated that both approaches significantly enhanced longitudinal safety on the expressway compared to the non-CV scenario. Further, Yang et al. [55] suggest that the utilization of connected vehicles could serve as a practical approach to reduce the occurrence of secondary crashes. Despite projections indicating an upsurge in the adoption of connected vehicle (CV) technology in the near future, it's important to recognize that a mixed fleet of both CV-equipped and non-CV-equipped vehicles will coexist for foreseeable future. As connected vehicles are introduced into the existing traffic mix, alongside vehicles lacking CV technology, they are not exempt from the potential risks of secondary collisions despite their enhanced communication abilities.

## Conclusions and summary

A novel approach to identify secondary crashes is investigated using temporal self-exciting point process model. The proposed approach is used to model crash events on I-4 highway in Florida, USA, spanning a period of three years. The model has the ability to analyze and classify crash events data using either a stationary background rate, a sinusoidal or a piecewise non-stationary background rates, for periodic variation in crash events data. After fitting the model to the data and optimizing the model parameters, queue time values resulting from primary crashes are determined and used to classify the crash events into primary and secondary events. The stationary background rate model failed to appropriately fit the data, since it is based on the assumption that the crash events are invariant to any external factors. However, the data from I-4 exhibits periodic variation with weekly and daily trends based on rush-hours period inherent to traffic on highways. Using non-stationary background rate models, we were able to accurately fit the crash data and obtain queue time curves with peak on Fridays and trough on Sundays similar to the crash data. Using the sinusoidal non-stationary background rate model, we find that 3.38% to 15.09% of the traffic incidents from the crash data on I-4 are secondary. The proposed models in this work can be used to create policies and countermeasures that aim to reduce the risk of secondary crashes.

## Supporting information

**S1 File.**
(ZIP)

## Author Contributions

**Conceptualization:** Samarth Motagi, Sirish Namilae, Audrey Gbaguidi.

**Data curation:** Samarth Motagi, Scott Parr.

**Formal analysis:** Samarth Motagi, Audrey Gbaguidi.

**Funding acquisition:** Sirish Namilae, Scott Parr.

**Investigation:** Samarth Motagi, Sirish Namilae.

**Methodology:** Samarth Motagi, Sirish Namilae, Audrey Gbaguidi.

**Project administration:** Sirish Namilae, Scott Parr, Dahai Liu.

**Resources:** Sirish Namilae.

**Software:** Samarth Motagi, Audrey Gbaguidi.

**Supervision:** Sirish Namilae, Scott Parr, Dahai Liu.

**Validation:** Samarth Motagi.

**Visualization:** Sirish Namilae.

**Writing – original draft:** Samarth Motagi, Audrey Gbaguidi.

**Writing – review & editing:** Sirish Namilae, Scott Parr, Dahai Liu.

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
