## [Decision Letter · Decision Letter 0]

15 Aug 2023

PONE-D-23-17534Point-Process Modeling of Secondary CrashesPLOS ONE

Dear Dr. Namilae,

Thank you for submitting your manuscript to PLOS ONE. After careful consideration, we feel that it has merit but does not fully meet PLOS ONE’s publication criteria as it currently stands. Therefore, we invite you to submit a revised version of the manuscript that addresses the points raised during the review process.

We look forward to receiving your revised manuscript.

Kind regards,

Yajie Zou

Academic Editor

PLOS ONE

“This research was funded by DOT-UTC Center for Advanced Transportation Mobility.”

4. We note that Figure 2 in your submission contain [map/satellite] images which may be copyrighted. All PLOS content is published under the Creative Commons Attribution License (CC BY 4.0), which means that the manuscript, images, and Supporting Information files will be freely available online, and any third party is permitted to access, download, copy, distribute, and use these materials in any way, even commercially, with proper attribution. For these reasons, we cannot publish previously copyrighted maps or satellite images created using proprietary data, such as Google software (Google Maps, Street View, and Earth). For more information, see our copyright guidelines: http://journals.plos.org/plosone/s/licenses-and-copyright.

1. You may seek permission from the original copyright holder of Figure 2 to publish the content specifically under the CC BY 4.0 license. 

5. We are unable to open your Supporting Information files. Please kindly revise as necessary and re-upload.

Reviewers' comments:

Reviewer's Responses to Questions

**Comments to the Author**

1. Is the manuscript technically sound, and do the data support the conclusions?

Reviewer #1: Partly

Reviewer #2: Partly

Reviewer #3: Partly

2. Has the statistical analysis been performed appropriately and rigorously? 

Reviewer #1: Yes

Reviewer #2: Yes

Reviewer #3: Yes

3. Have the authors made all data underlying the findings in their manuscript fully available?

Reviewer #1: Yes

Reviewer #2: Yes

Reviewer #3: Yes

4. Is the manuscript presented in an intelligible fashion and written in standard English?

Reviewer #1: Yes

Reviewer #2: Yes

Reviewer #3: Yes

5. Review Comments to the Author

Reviewer #1: Normally, after a primary traffic accident is reported. A nearby traffic incident is classified as the secondary traffic accident. What is the practical contribution of this study?

Some ground truth data should be collected to evaluate the performance of the proposed method.

Some state-of-the-art methods should be considered as the benchmark models to validate the proposed method.

Some typos in the study should be corrected and the manuscript should be carefully proofread.

With the development of the CAV technology, some traffic platooning strategies can be used to improve the traffic safety. The authors are suggested to review this field and discuss in the manuscript. For example, see: Operation analysis of freeway mixed traffic flow based on catch-up coordination platoon. Accident Analysis & Prevention. Longitudinal safety evaluation of connected vehicles’ platooning on expressways. Accident Analysis & Prevention.

Reviewer #2: Overall, the manuscript is a well-organized paper, which proposed a new approach to identify secondary crash and evaluated with real-word dataset and compared with other existing approaches. Here are several comments/questions.

1. It seems the model was trained with 2015-2017 data to determine the parameters, it would be better to fit with the most recent data, if the data is available.

2. For the model used in the manuscript, it works well with earthquake, but it may not work perfectly with crash data. The model seems only consider the time dimension of crash events, but it does not involve the spatial attribute of crash events. Instead, it assumes any event within 2 miles and obtained queue time from model is the secondary crash. The result from this approach can be quite similar with using static spatial and temporal threshold, for example 120 minutes and 2 miles. Also, when you using the 2 miles, are you only considering the crash happening within 2 miles downstream or also includes the crashes upstream?

3. The manuscript seems only attempts to identify the crash queue time, which is at the city level, but the queue time could be quite different in different locations and sites. With the overall queue time, it may cause false positives for some secondary crashes.

4. In the formulas 8 and 9, it seems the weekly trend uses the sin function, while the rush hour trend uses the cos function. Why are they different and how do you derive these formulas?

Minor suggestions:

1. Line 42-44, it would be better to update these facts with recent (2022 or 2023) statistics.

2. Line 79-82, the sentences are not clear, need to be rephrased.

Reviewer #3: 1. How to define the secondary crash in this study?

2. What are the main purposes for analyzing the stationary background and non-stationary background?

3. In the discussion section, why not compare and summarize the accuracy of identifying secondary crashes from other studies?

4. Some recent studies in this field should be reviewed.

5. How to validate the accuracy of the proposed secondary crash identification approach.

6. PLOS authors have the option to publish the peer review history of their article (what does this mean?). If published, this will include your full peer review and any attached files.

Reviewer #1: No

Reviewer #2: No

Reviewer #3: No

---

## [Author Response · Author response to Decision Letter 0]

12 Sep 2023

The detailed response to reviewers comments is uploaded in the file "Reviewer comments".

---

## [Decision Letter · Decision Letter 1]

21 Nov 2023

Point-Process Modeling of Secondary Crashes

PONE-D-23-17534R1

Dear Dr. Namilae,

We’re pleased to inform you that your manuscript has been judged scientifically suitable for publication and will be formally accepted for publication once it meets all outstanding technical requirements.

Kind regards,

Ahmed Mancy Mosa, Ph.D.

Academic Editor

PLOS ONE

Additional Editor Comments (optional):

Reviewers' comments:

Reviewer's Responses to Questions

**Comments to the Author**

1. If the authors have adequately addressed your comments raised in a previous round of review and you feel that this manuscript is now acceptable for publication, you may indicate that here to bypass the “Comments to the Author” section, enter your conflict of interest statement in the “Confidential to Editor” section, and submit your "Accept" recommendation.

Reviewer #3: All comments have been addressed

2. Is the manuscript technically sound, and do the data support the conclusions?

Reviewer #3: (No Response)

3. Has the statistical analysis been performed appropriately and rigorously? 

Reviewer #3: (No Response)

4. Have the authors made all data underlying the findings in their manuscript fully available?

Reviewer #3: (No Response)

5. Is the manuscript presented in an intelligible fashion and written in standard English?

Reviewer #3: (No Response)

6. Review Comments to the Author

Reviewer #3: (No Response)

7. PLOS authors have the option to publish the peer review history of their article (what does this mean?). If published, this will include your full peer review and any attached files.

Reviewer #3: No

---

## [Editor Report · Acceptance letter]

4 Dec 2023

PONE-D-23-17534R1 

Point-Process Modeling of Secondary Crashes 

Dear Dr. Namilae:

I'm pleased to inform you that your manuscript has been deemed suitable for publication in PLOS ONE. Congratulations! Your manuscript is now with our production department. 

Kind regards, 

on behalf of

Dr. Ahmed Mancy Mosa 

Academic Editor

PLOS ONE